# Effects of timing of cord clamping on neonatal hemoglobin and bilirubin levels in preterm and term infants—A prospective observational cohort study

Shikha Malik[1☉], Madhuri Kapu[1☉], Mahendra Kumar Jain[2]*, Bhupeshwari patel[3], Nandkishor kabra[4]

1 Department of Pediatrics, AIIMS Bhopal, Bhopal, Madhya Pradesh, India, 2 Department of Neonatology, AIIMS Bhopal, Bhopal, Madhya Pradesh, India, 3 Department of Trauma and Emergency medicine, AIIMS Bhopal, Bhopal, Madhya Pradesh, India, 4 Surya hospital, Mumbai, Maharashtra, India

☉ These authors contributed equally to this work.
* mahendra.neonatology@aiimsbhopal.edu.in

**Data Availability Statement:** Data used for this study are available from the AIIMS Bhopal Institutional Human Ethics Committee-PostGraduate Research (IHEC-PGR) to researchers

## Abstract

### Background

Delayed cord clamping (DCC) is a proven beneficial intervention, but the suggested timings of DCC vary from 30 to 300 seconds after birth or until cord pulsation stops. This study aimed to find the optimum timing of DCC to maximize the benefits such as an increase in hemoglobin, and hematocrit without increasing the risks of polycythemia and hyperbilirubinemia.

### Methods

We conducted a single-center prospective observational cohort study. All singleton neonates with gestational age $\geq$ 28 weeks born at the center in the 17 months of the study period from November 2020 to March 2022 were enrolled. Participants were divided into four groups based on DCC time: group A: <60 sec, group B: 60–119 sec, group C: 120–180 sec, and group D: >180 sec. The primary outcome was the levels of hemoglobin, hematocrit, and bilirubin at 48 hours of life.

### Results

Four hundred and eight neonates were enrolled. They were divided into four groups based on the timing of DCC (group A: n = 52, group B: n = 137, group C: n = 155, group D: n = 64). With an increase in the duration of DCC, there was an increase in the level of hemoglobin and hematocrit without an increase in the risk of polycythemia or neonatal hyperbilirubinemia. The benefits were best in group C (120–180 sec) and group D (>180 sec).

who meet the criteria for access to confidential data at ihec@aiimsbhopal.edu.in. Further, interested researchers may be asked to send appropriate requests to the corresponding author at mahendra.neonatology@aiimsbhopal.edu.in.

**Funding:** The author(s) received no specific funding for this work.

**Competing interests:** The authors have declared that no competing interests exist.

## Conclusions

DCC of $\geq$ 120 seconds appears to be optimal where hemoglobin and hematocrit are highest without an increase in the risk of neonatal hyperbilirubinemia. The risk of adverse effects like polycythemia or neonatal hyperbilirubinemia requiring phototherapy did not increase even after extending the time of cord clamping to >180 seconds.

## Introduction

Delayed cord clamping (DCC) is a high-impact, low-cost, evidence-based recommended intervention in preterm and term newborn infants. On the other hand, the optimal timing of DCC has always been a hotly debated issue. The recommended timing for DCC varies between the guidelines: World Health Organization (2014) recommends DCC over 1–3 minutes. Neonatal Resuscitation Protocol (NRP, 8th Edition, 2020) by the AAP/AHA and American College of Obstetrics and Gynecology (ACOG) recommends DCC at 30–60 sec. National Institute for Health and Care Excellence (NICE, 2017) recommends DCC over 1 minute or longer if the mother requests, International Federation of Obstetrics and Gynecology recommends DCC at 30 seconds in preterm infants <34 weeks of gestational age and 30 seconds to 3 minutes in term infants [1–5]. DCC in full-term neonates leads to increased iron stores, higher hematocrit, and polycythemia. However, it does not increase the risk of symptomatic polycythemia, or hyperbilirubinemia requiring phototherapy [6]. DCC is associated with short-term benefits in preterm neonates, including reduced incidence of intraventricular hemorrhage, necrotizing enterocolitis, sepsis, and mortality. Moreover, the additional short-term benefits of DCC in preterm infants include: an increased hematocrit at birth, decreased risk of anemia, decreased need for blood transfusions, and better long-term neuro-developmental outcomes [2,6]. There has always been a theoretical concern about the increased risk of hypothermia with a longer duration of DCC. Neonatal hypothermia, directly and indirectly, contributes to an increased risk of neonatal morbidities and mortality [7]. The greatest risk of hypothermia occurs within the first few minutes of life because of the wide variation in environmental temperature from intrauterine life to extrauterine life [8]. It has been observed that the temperature of lambs was better maintained with DCC than with conventional care and immediate cord clamping [9].

We hypothesized that increasing the duration of cord clamping might improve early hematological parameters without any untoward effects. The present observational study was conducted to determine the optimal time of DCC clamping in preterm and term infants that maximize benefits without increasing the risk of polycythemia or neonatal hyperbilirubinemia.

## Materials and methods

This was a single-center; prospective observational study conducted between the period of November 2020—March 2022 at the Department of Pediatrics and Neonatology at All India Institute of Medical Sciences, Bhopal. The primary outcome was the levels of hemoglobin, hematocrit, and bilirubin at 48 hours of life. All neonates with gestational age $\geq$ 28 weeks born either by vaginal route or lower segment cesarean section (LSCS) were considered eligible for the study. Following neonates were excluded from the study: families not willing to participate, infants born to mothers with clinical diseases (gestational diabetes, hypertensive disorders of pregnancy), and complications of pregnancy (polyhydramnios, oligohydramnios, placental previa, abruptio placenta). Infants with hemolytic diseases and other diseases affecting

bilirubin metabolism, multiple gestations, and those requiring resuscitation at birth were also excluded from the study. Informed written consent for the study was obtained from the pregnant mother for enrolment into the study after explaining the purpose of the study. They were also provided with a participant information sheet.

Before embarking on the study the obstetric team was consulted and the importance of DCC was elaborated. They were requested to practice DCC for at least ≥30 seconds, discretion in a personal matter. However, the obstetrician team is not aware of the purpose of the study and the 4 groups mentioned in this study. They were requested to practice DCC for at least ≥30 seconds, discretion in a personal matter. We only did study on spontaneously breathing infants. The neonate who was not able to breathe at birth or required any type of resuscitation at birth as per the NRP 2020 guidelines was not included in the study group.

After delivery, neonates were positioned over the mother's abdomen (for babies delivered through the vaginal route) or between the mother's thighs (for babies delivered by LSCS). The precise time of DCC was noted as follows: from the time the baby is completely delivered to the time after which the obstetrician does the cord clamping using a digital stop clock accurate to 1 second. Based on timing of cord clamping the participants were divided on four groups: group A: <60 sec, group B: 60–119 sec, group C: 120–180 sec, and group D: > 180 sec. As the study is a descriptive cohort study, matching is not done. During this time, newborns were provided with routine post-birth care and observed for APGAR scores at 1 and 5 minutes.

According to our institutional protocol, oxytocin was given at birth after infants was born but prior to cord clamping after delivery. Oxytocin was given intramuscularly (to mothers who delivered by vaginal route) or intravenously (to mothers who were delivered by LSCS). All neonates were managed subsequently as per our standard Institutional NICU protocol. At 48 hours of life, one ml of venous blood sample was collected by venipuncture from a peripheral vein on the dorsal aspect of any hand for complete blood count (CBC: Hemoglobin—Hb, Hematocrit—HCT) and total serum bilirubin (TSB). CBC was analyzed by an automated hematology analyzer [Mindray, Sysmex] in the central lab. TSB values were analyzed by Beckman Coulter AU 680 chemistry analyzer in the central lab. Information regarding an infant who had hyperbilirubinemia and fulfilled the criteria for treatment with phototherapy (AAP Guidelines) was recorded. Polycythemia was defined as a hematocrit value of (HCT > 65).

Baseline maternal and neonatal data and laboratory reports were recorded in the proforma sheet. All data about the participants were available in the hospital records and could be retrieved by the patient identification number which is the unique health identification number (UHID number). All the data were uploaded into the Excel Sheet (Microsoft Excel 2008) and analysis was performed using SPSS version 12.0. Baseline and outcome data were provided as absolute frequencies and percentages for qualitative variables and as mean ± standard deviation or median with interquartile range for quantitative variables as appropriate. We used the Chi-square test or Fisher's exact test as appropriate to compare qualitative variables. Comparison of numerical data across four study groups was carried out by using One-way ANOVA (Parametric ANOVA or Non-parametric Kruskal-Wallis ANOVA) depending on the distribution of data. Post-hoc pairwise tests were performed using the Dunn test with Sidak correction for inter-group comparison. A $p$-value of $< 0.05$ was considered statistically significant. The study was approved by the Institutional Ethical Committee (Ref Number- 2020/PG/Jan/18, Date—20th November 2020). Sample size calculation was not done as time bound convenient sample was collected.

## Results

Fig 1 depicts the flow diagram of participant recruitment in this study. A total of 753 vaginal and LSCS deliveries were performed during the study period. After exclusions, a total of 408

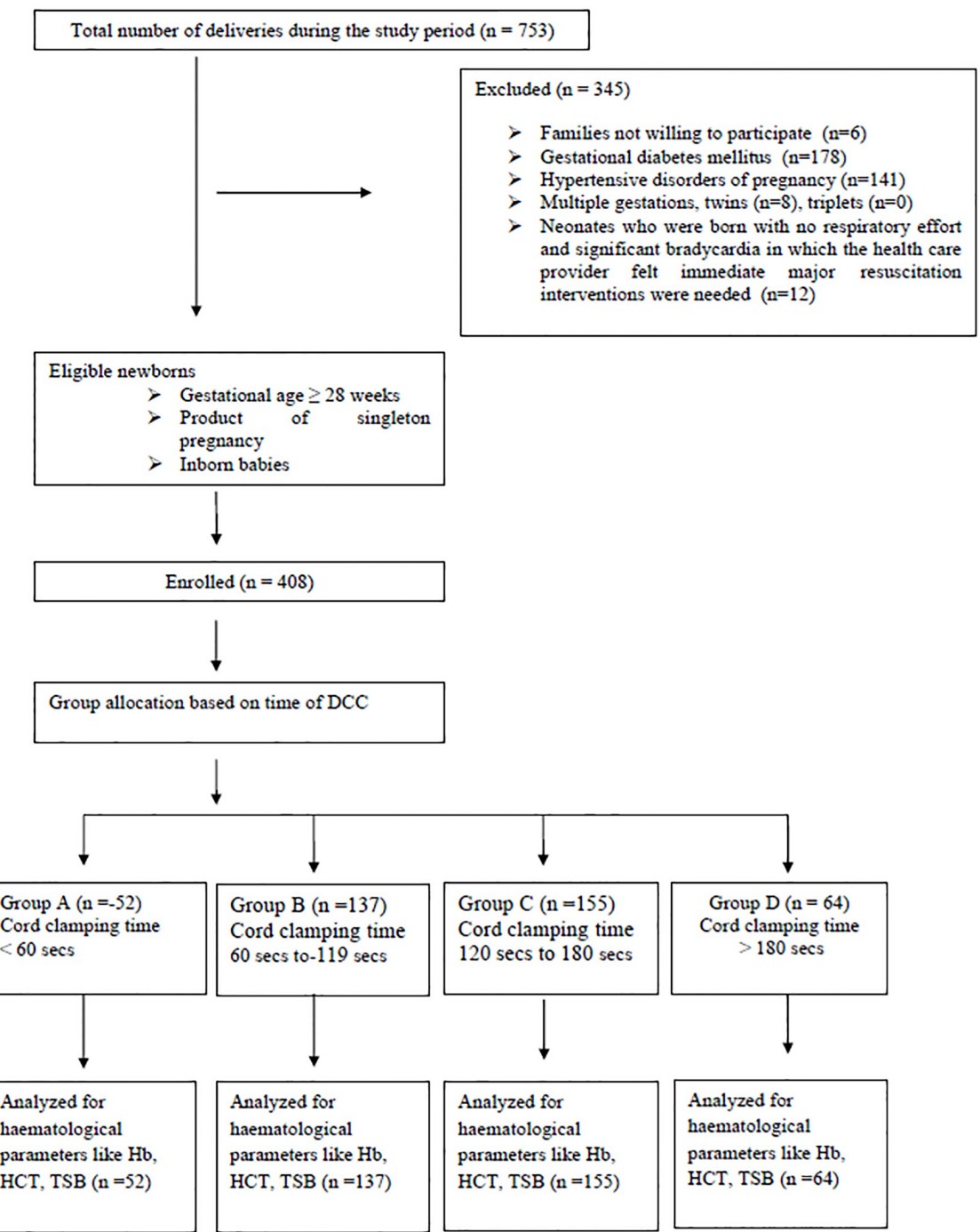

**Fig 1. Participant flow diagram.**

neonates, who met the inclusion criteria were recruited. Participants were divided into 4 groups based on cord clamping time following delivery of newborn: group A: <60 sec, group B: 60–119 sec, group C: 120–180 sec, and group D: > 180 sec. Out of these 408 neonates, 353 were term and 55 were preterm newborns. The birth weight distribution of study infants was as follows: ≥ 2500 g (n = 329), 1500 to 2499 g (n = 79), 1000 to 1499 g (n = 6), and <1000g

**Table 1. Baseline characteristics of study participants*.**

| Baseline Characteristics | Timing of DCC | | | | p value |
|---|---|---|---|---|---|
| | Group A <br> <60s <br> (n = 52) | Group B <br> 60-119s <br> (n = 137) | Group C <br> 120-180s <br> (n = 155) | Group D <br> >180s <br> (n = 64) | |
| Time of cord clamping (s) | 27.31 ± 10.82 | 78.12 ± 15.00 | 143.42 ± 25.01 | 220.81 ± 54.77 | <0.001# |
| Gestational Age (weeks) | 38.12 ± 2.51 | 38.02 ± 2.01 | 38.08 ± 2.02 | 38.53 ± 1.62 | 0.396# |
| Gravida | | | | | 0.890$ |
| G1 | 11(39.3%) | 43(42.2%) | 50(44.2%) | 21(51.2%) | |
| G2 | 9 (32.1%) | 33 (32.4%) | 39(34.5%) | 13 (31.7%) | |
| ≥ G3 | 8 (28.6%) | 26(25.5%) | 24 (21.2%) | 7 (17.1%) | |
| Mode of Delivery | | | | | 0.073$ |
| NVD | 21 (40.4%) | 69(50.4%) | 93 (60.0%) | 36 (56.2%) | |
| LSCS | 31(59.6%) | 68 (49.6%) | 62 (40.0%) | 28 (43.8%) | |
| Gender | | | | | 0.866$ |
| Male | 27 (51.9%) | 73 (53.3%) | 89 (57.4%) | 35 (54.7%) | |
| Female | 25 (48.1%) | 64 (46.7%) | 66 (42.6%) | 29 (45.3%) | |
| Mothers HB (g/dl) | 12.32 ± 1.36 | 12.14 ± 1.38 | 12.58 ± 1.34 | 13.32 ± 0.82 | <0.001# |
| Weight for Gestational Age | | | | | 0.154$ |
| AGA | 36 (69.2%) | 106 (77.4%) | 121 (78.1%) | 41 (64.1%) | |
| SGA | 16 (30.8%) | 31 (22.6%) | 32 (20.6%) | 23 (35.9%) | |
| LGA | 0(0.0%) | 0 (0.0%) | 2 (1.3%) | 1 (0.0%) | |
| Birth Weight | 2.64 ± 0.46 | 2.73 ± 0.45 | 2.79 ± 0.40 | 2.79 ± 0.27 | 0.105# |

(# One way parametric ANOVA, $ Fisher's Exact Test).

*supporting information "Table 1 doc".

(n = 1). The gestational age distribution of study infants was as follows: ≥ 37 weeks (n = 353), 32 to 36+6 weeks (n = 52), and 28 to 31+6 weeks (n = 1); Baseline characteristics of enrolled newborns are summarized in Table 1. Study outcomes across the four study groups are summarized in Table 2.

With an increase in the duration of DCC, there was an increase in the level of Hemoglobin and Hematocrit without an increase in the risk of polycythemia or neonatal hyperbilirubinemia. There was a significant difference between the 4 groups in terms of baby Hb (g/dL); p <0.001. The median baby Hb being highest in the DCC group D (>180s), the strength of association (Kendall's Tau) = 0.15 (small effect size). However, when compared between all 4 groups, Hb levels were statistically significantly higher when DCC was performed between 120 and 180 seconds compared to the group <60 seconds and the group 60–120 seconds (adjusted P value < 0.001). There were no significant differences in Hb levels between the two groups 120–180 sec and >180 sec (adjusted p-value 0.943). The HCT was statistically highest in the cord clamping group >180 sec. Bilirubin levels were not significantly different in the four study groups.

## Discussion

The Hb and HCT values increased with an increase in the duration of DCC. A higher statistically significant incremental increase in Hb and HCT was observed in the DCC groups of 120–180 seconds and 180 seconds. There was no significant increase in the risk of polycythemia or hyperbilirubinemia requiring phototherapy with an increase in the duration of DCC. A

**Table 2. Comparison of neonatal outcomes in the study groups\*.**

| Outcomes | Timing of DCC | | | | Overall p value across all four groups | Adjusted p value Dunn test with Sidak correction for inter group comparison |
|---|---|---|---|---|---|---|
| | Group A <60s (n = 52) | Group B 60-119s (n = 137) | Group C 120-180s (n = 155) | Group D >180s (n = 64) | | |
| Baby Hemoglobin (g/dL) Median (IQR) | 17.40 (15.35 to 20.02) | 19.10 (16.80to 20.80) | 19.20 (18.05 to 20.75) | 19.50 (18.40 to 20.50) | <0.001 [#] | A vs. B (p = 0.009) A vs. C (p < 0.001) A vs. D (p < 0.001) B vs. C (p < 0.001) B vs. D (p = 0.344) C vs. D (p = 0.943) |
| HCT (%) Median (IQR) | 50.95 (44.68 to 55.62) | 52.00 (48.9 to 58.4) | 54.50 (50.9 to 58.45) | 58.90 (56.70 to 61.50) | <0.001 [#] | A vs. B (p = 0.040) A vs. C (p < 0.001) A vs. D (p < 0.001) B vs. C (p < 0.001) B vs. D (p < 0.001) C vs. D (p < 0.001) |
| Total Serum Bilirubin (mg/dL) Median (IQR) | 9.70 (8.28 to 10.93) | 9.80 (7.78 to 11.9) | 8.91 (6.28 to 11.73) | 9.90 (7.70 to 12.07) | 0.296 [#] | A vs. B (p = 1) A vs. C (p = 0.998) A vs. D (p = 0.845) B vs. C (p = 1) B vs. D (p = 0.484) C vs. D (p = 0.317) |
| Hyperbilirubinemia requiring phototherapy | 2 (3.8%) | 9 (6.6%) | 10 (6.5%) | 2 (3.1%) | 0.782 [$] | - - |
| Polycythemia (HCT > 65) | 0(0%) | 8 (5.8%) | 2 (1.3%) | 2 (3.1%) | 0.076 [$] | - - |

\*supporting information "Table 2 docx".

limitation of our study was that it was not a randomized controlled trial but a prospective observational cohort study. Therefore all the inherent limitations of observational studies are applicable. The outcome variables of Hb, HCT, and bilirubin were not normally distributed in the study groups and we needed to perform a non-parametric Kruskal Wallis ANOVA test to make overall comparisons. Findings similar to our study were also noted in previously published studies [9–12,14]. Interestingly, there were no significant differences in hemoglobin levels between the two groups 120–180 sec and >180 sec (adjusted p-value 0.943). The hematocrit was statistically highest in the cord clamping group >180 sec compared to the cord clamping group 120–180 sec (adjusted p-value <0.001). In our study, there was no significant difference among the four groups in the incidence of neonatal hyperbilirubinemia requiring phototherapy and polycythemia. In contrast, in another study, there was a trend towards a higher risk of neonatal jaundice requiring phototherapy when cord clamping was performed for 90–120seconds [13]. A randomized control trial from Nepal that studied the long-term effects of delayed cord clamping (≥180 s) compared to early cord clamping (≤60 s) on the neurodevelopmental outcome at 3 years of age by Ages and Stages Questionnaire (ASQ) found no significant differences in outcomes between two groups [14]. Adequately powered large randomized controlled trial on the preferred time duration of DCC is urgently required that assesses short and long-term benefits.

## Conclusions

DCC of >120 seconds appears to be optimal where hemoglobin and hematocrit are highest without an increase in the risk of polycythemia or neonatal hyperbilirubinemia. The risk of

adverse effects like neonatal hyperbilirubinemia requiring phototherapy or polycythemia did not increase even after extending the time of cord clamping to >180 seconds.

## Supporting information

**S1 Data.**
(DOC)

## Author Contributions

**Conceptualization:** Shikha Malik, Mahendra Kumar Jain, Bhupeshwari patel.

**Data curation:** Madhuri Kapu, Bhupeshwari patel.

**Formal analysis:** Shikha Malik, Madhuri Kapu, Nandkishor kabra.

**Funding acquisition:** Bhupeshwari patel.

**Investigation:** Shikha Malik, Madhuri Kapu, Mahendra Kumar Jain.

**Methodology:** Shikha Malik, Mahendra Kumar Jain, Nandkishor kabra.

**Resources:** Bhupeshwari patel.

**Supervision:** Shikha Malik, Mahendra Kumar Jain, Bhupeshwari patel.

**Validation:** Shikha Malik, Bhupeshwari patel, Nandkishor kabra.

**Writing – original draft:** Mahendra Kumar Jain.

**Writing – review & editing:** Shikha Malik.

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
