## [Decision Letter · Decision Letter 0]

22 Aug 2023

PONE-D-23-08062Effects of Timing of Cord Clamping on Neonatal Hemoglobin and Bilirubin Levels in Preterm and Term Infants – A Prospective Observational Cohort Study

Dear Dr. Mahendra Kumar Jain,

Thank you for submitting your manuscript to PLOS ONE. After careful consideration, we feel that it has merit but does not fully meet PLOS ONE’s publication criteria as it currently stands. Therefore, we invite you to submit a revised version of the manuscript that addresses the points raised during the review process.

We look forward to receiving your revised manuscript.

Kind regards,

Sanjoy Kumer Dey, M.D

Academic Editor

PLOS ONE

Reviewers' comments:

Reviewer's Responses to Questions

**Comments to the Author**

1. Is the manuscript technically sound, and do the data support the conclusions?

Reviewer #1: Yes

Reviewer #2: Partly

2. Has the statistical analysis been performed appropriately and rigorously? 

Reviewer #1: Yes

Reviewer #2: Yes

3. Have the authors made all data underlying the findings in their manuscript fully available?

Reviewer #1: Yes

Reviewer #2: Yes

4. Is the manuscript presented in an intelligible fashion and written in standard English?

Reviewer #1: Yes

Reviewer #2: Yes

5. Review Comments to the Author

Reviewer #1: I congratulate team for conducting such good study. Sample size is adequate and this study provide new insight about the optimal time of cord clamping as DCC is being recommended for all term newborn,

Reviewer #2: Thank you for the opportunity to review this single-centre observational cohort study of infants ≥28 weeks’ gestation at birth. The study has limitations in what it can add to knowledge in this area. There are previous larger studies addressing similar questions that have more rigorous design. There is certainly some interest in determining what the ‘correct’ duration of cord clamping is, although more recent RCTs have focused on the individual response of infant, acknowledging that arbitrary time-based strategies (whether in two categories or four, as here) don’t really make physiological sense. More critically, I think there are a number of elements that should reasonably be expected from a well-designed cohort study that are missing here. There are important outcomes that have been omitted that are really necessary to properly interpret the data. My detailed comments are below:

“The present observational study was conducted to determine the optimal time of DCC

clamping in preterm and term infants that maximize benefits without increasing the risk of

polycythemia or neonatal hyperbilirubinemia.”

This aim is fairly broad. Was there any specific pre-specified hypothesis?

Methods

“Before embarking on the study the obstetric team was consulted and the importance of DCC

was elaborated. They were requested to practice DCC of at least ≥30 seconds at their

discretion in an individual case.”

I am curious to know if this is the total extent of the advice given. Were the obstetric team aware that the purpose of the study was to compare groups based on time to cord clamping after birth? Or given any other advice on how much beyond 30 seconds they should target? There is fairly substantial risk of bias here, as there may have been other clinical factors that influenced the obstetrician’s chosen DCC time, which could also impact outcomes. If the obstetricians were aware of the study design, they could (consciously or sub-consciously) have adjusted their approach to time of cord clamping accordingly.

“According to our institutional protocol, after delivery, Oxytocin was given”

To clarify: was oxytocin given after the infant was born but prior to cord clamping, or after both birth and cord clamping were complete.

Was the study prospectively registered in a clinical trials registry? Has a STROBE checklist been prepared?

Can the authors provide a copy of the data collection sheet used for the study? Were any other data points collected beyond those included in the baseline data table and the laboratory test results? I ask this because there are specific outcomes that I would consider important to collect based on the study aim, which is to identify the DCC time that can “maximise benefits without increasing risk of polcythemia or hyperbilirubinaemia”. These last two are addressed, but the benefits are not. The authors mention several relevant outcomes in the introduction: temperature/hypothermia, IVH, NEC, sepsis, mortality, need for transfusion, neurodevelopment. The last outcome is unlikely to feasible, but the others would be. The study aim cannot be achieved if none of these were assessed.

It does not appear that any primary outcome was identified a priori – is this correct?

Was there any specific reason for the time frame chosen for the study? It could still have been possible to estimate how well the study was powered if a primary outcome had been chosen and an estimate of patient numbers was available.

There are a number of preclinical studies indicating that the early physiological benefit of DCC is not dependent upon a fixed time period, but on whether lung aeration has been achieved prior to cord clamping (see Bhatt 2013, Polglase 2015, and others). Clinical studies have been conducted utilizing this approach, ‘physiological-based cord clamping’ (work by Knol, Brouwer, Badurdeen, the recently completed ABC3 trial and VentFirst trials). I would expect this concept to have been discussed by the authors. It would be interesting to know more about the time of regular spontaneous breathing in this population, or even just the respiratory component of the APGAR scores. This is one reason I asked about other data points that were collected above.

6. PLOS authors have the option to publish the peer review history of their article (what does this mean?). If published, this will include your full peer review and any attached files.

Reviewer #1: **Yes: **Dr Deepak Kumar Sharma

Reviewer #2: No

---

## [Author Response · Author response to Decision Letter 0]

9 Nov 2023

all required documents with standard format attached

---

## [Editor Report · Decision Letter 1]

4 Dec 2023

Effects of Timing of Cord Clamping on Neonatal Hemoglobin and Bilirubin Levels in Preterm and Term Infants – A Prospective Observational Cohort Study

PONE-D-23-08062R1

Dear Dr. Jain,

We’re pleased to inform you that your manuscript has been judged scientifically suitable for publication and will be formally accepted for publication once it meets all outstanding technical requirements.

Kind regards,

Sanjoy Kumer Dey, M.D

Academic Editor

PLOS ONE
---

## [Editor Report · Acceptance letter]

19 Dec 2023

PONE-D-23-08062R1 

PLOS ONE

Dear Dr. Jain, 

I'm pleased to inform you that your manuscript has been deemed suitable for publication in PLOS ONE. Congratulations! Your manuscript is now being handed over to our production team.

Kind regards, 

on behalf of

Dr. Sanjoy Kumer Dey 

Academic Editor

PLOS ONE